# OwlFusion: Depth-Only Onboard Real-Time 3D Reconstruction of Scalable Scenes for Fast-Moving MAV

**Guohua Gou** **, Xuanhao Wang, Haigang Sui \*, Sheng Wang, Hao Zhang and Jiajie Li**

State Key Laboratory Information Engineering in Surveying, Mapping and Remote Sensing, Wuhan University, Wuhan 430070, China; guohua.gou@whu.edu.cn (G.G.); xuanh.w@whu.edu.cn (X.W.); wsheng@whu.edu.cn (S.W.); zhanghao.1003@whu.edu.cn (H.Z.); 2017301200143@whu.edu.cn (J.L.)

\* Correspondence: 00201543@whu.edu.cn

**Abstract:** Real-time 3D reconstruction combined with MAVs has garnered significant attention in a variety of fields, including building maintenance, geological exploration, emergency rescue, and cultural heritage protection. While MAVs possess the advantages of speed and lightness, they also exhibit strong image blur and limited computational resources. To address these limitations, this paper presents a novel approach for onboard, depth-only, real-time 3D reconstruction capable of accommodating fast-moving MAVs. Our primary contribution is a dense SLAM system that combines surface hierarchical sparse representation and particle swarm pose optimization. Our system enables the robust tracking of high-speed camera motion and facilitates scaling to large scenes without being constrained by GPU memory resources. Our robust camera tracking framework is capable of accommodating fast camera motions and varying environments solely by relying on depth images. Furthermore, by integrating path planning methods, we explore the capabilities of MAV autonomous mapping in unknown environments with restricted lighting. Our efficient reconstruction system is capable of generating highly dense point clouds with resolutions ranging from 2 mm to 8 mm on surfaces of different complexities at rates approaching 30 Hz, fully onboard a MAV. We evaluate the performance of our method on both datasets and real-world platforms and demonstrate its superior accuracy and efficiency compared to existing methods.

**Keywords:** onboard 3D reconstruction; micro aerial vehicles (MAVs); RGB-D SLAM; fast camera tracking; particle swarm pose optimization; hierarchical sparse 3D representation

## 1. Introduction

With the rapid development and extensive utilization of micro aerial vehicles (MAVs), MAVs equipped with online 3D reconstruction capabilities have gained considerable attention and importance. By autonomously capturing 3D information about the environment to optimize their action path and behavioral decisions, MAVs can enhance the efficiency and precision of task execution in complex environments. Additionally, they can achieve environment modeling and map construction without human intervention, relieving humans from repetitive or hazardous tasks. Hence, MAVs have the potential to be widely employed in areas such as building maintenance, geological exploration, and emergency rescue. The research on simultaneous localization and dense mapping (Dense SLAM) methods has been advancing with the progression of computing power. Powerful *Graphics Processing Units* (GPUs) are now widely accessible, enabling vision algorithms to process large quantities of data in real-time through parallel processing. Among various approaches in this field, KinectFusion [1] is one of the most representative techniques that enables the construction of dense 3D scenes in real-time via commodity depth sensors.

To apply dense SLAM in MAVs and adapt to challenging environments such as light-constrained settings during emergency rescue missions, two challenges must be addressed. Firstly, it is a challenge to robustly track fast camera motion, which results in significant

motion blur in RGB images, particularly in lighting-constrained conditions, and this issue is particularly significant for agile MAVs. Secondly, many existing systems [2–4] are limited by computational power, making it difficult to demonstrate their real-time capabilities on the onboard computer.

We propose OwlFusion, an end-to-end solution for onboarding dense RGB-D reconstruction of scalable scenes for fast-moving MAVs. Our method relies solely on depth information as the depth image actively sensed is not limited by lighting. Additionally, depth images typically do not produce the same full-frame pixel blur as RGB images when the camera is moving fast [4]. To address the challenges of dense SLAM application in fast-moving MAVs, we introduce two key methods. Firstly, we propose a fast pose estimation method that reduces the consumption of many unnecessary particle fitness evaluations by introducing planar constraints. This method accelerates the convergence efficiency of optimized iterations using the same computational resources and enables real-time tracking of high-speed camera motion on onboard computing devices. Secondly, we integrate surface hierarchical sparse representation and particle swarm pose optimization methods, and weighted depth fusion of depth images with noise at different voxel levels to achieve real-time, high-quality reconstruction of large-scale scene 3D models on an airborne computer with very limited computational and storage resources. We mounted a commodity depth camera, Intel RealSense D435i, on a MAV with a wheelbase of only 450 mm and achieved real-time 3D reconstruction of scalable scenes, as shown in Figure 1.

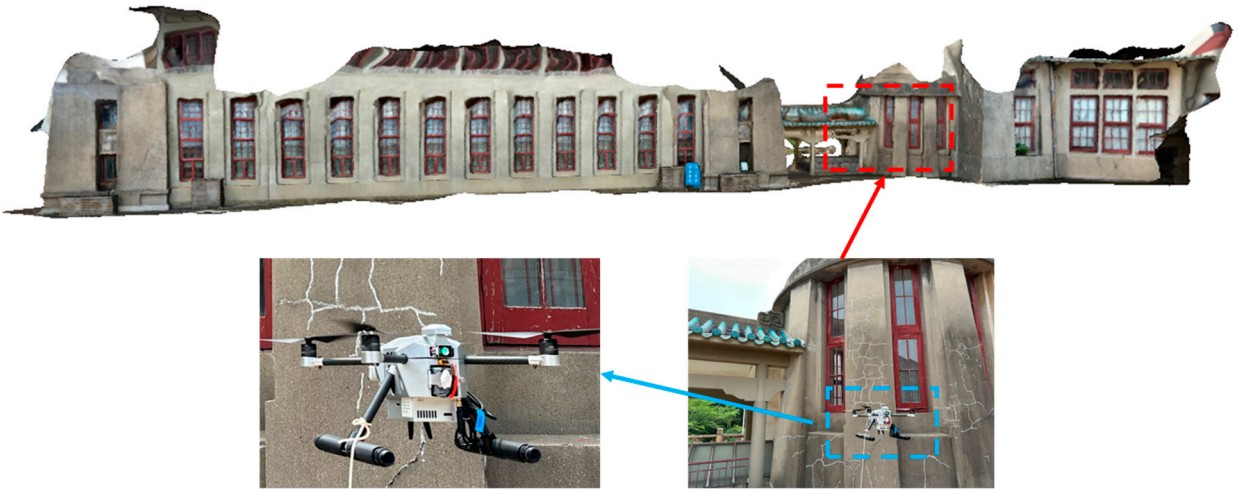

**Figure 1.** MAV performing onboard scalable scan and dense mesh being generated in real time. Our lightweight system is capable of generating sub-centimeter resolution meshes at 30 Hz, fully onboard.

This paper is divided into five sections. We have introduced the research issues and objectives in this section. Section 2 presents an overview of existing methods and their limitations, as well as a discussion of the relevant literature. Section 3 outlines the proposed methodology, with a general overview of the approach and detailed descriptions of individual building blocks in subsections. Section 4 reports on the experimental results, including a description of the setup, presentation of results, and analysis. Section 5 concludes the article with a summary of the findings, implications of the results, limitations of the study, and suggestions for future research.

## 2. Related Work

With advancements in GPU architecture and *General-Purpose Graphics Processing Unit* (GPGPU) algorithms, real-time 3D reconstruction has significantly improved since the inception of DTAM [5]. There has been a substantial increase in the number of publications in this field in the past decade. Unlike DTAM, which achieves real-time dense 3D scene reconstruction using RGB cameras, we are primarily interested in using low-cost RGB-

D sensors. These sensors are more readily available and provide more accurate depth information. In this section, we provide an overview of related systems, with a focus on 3D reconstruction methods that rely on RGB-D cameras and tracking methods for fast-moving cameras.

### 2.1. Real-Time RGB-D Reconstruction

The representative work for real-time dense 3D reconstruction using RGB-D cameras is KinectFusion [1]. It was the first to demonstrate convincing real-time 3D reconstruction results. Prior to this, there were also some great attempts, a famous example being the method of Curless and Levoy [6], which is based on active triangulation sensors such as laser range scanners and structured light cameras, and can generate very high-quality results. The characteristic of this method is the use of a fully volumetric data structure to implicitly store samples of the continuous function, with the depth map being transformed into a *Truncated Signed Distance Function* (TSDF), and the accumulated averages becoming a regular voxel grid. Finally, the reconstructed surface is extracted as the zero-level set of the implicit function through ray casting. Unfortunately, due to computing limitations, Curless and Levoy's work did not achieve the real-time 3D reconstruction of scene objects. KinectFusion inherits and improves upon such methods and achieves small-scale real-time dense 3D reconstruction with the help of high-performance graphics computing units.

To reconstruct larger spaces, Whelan et al. [2] extended the pipeline of KinectFusion, allowing voxels to flow out of the GPU based on camera motion to make room for storing new data. However, this moving volume process is one-way and lossy, with surface data being compressed into meshes, which cannot flow back to the GPU once it has moved to the CPU. To address this issue, Nießner et al. [7] proposed a bidirectional data exchange method that allows for the flexible exchange of reconstructed surface data between the GPU and CPU, pushing the application of real-time 3D reconstruction to even larger spaces. However, even with the support of the moving volume method and data exchange method, the use of regular voxels is still limited in reconstructing larger scenes. This is because regular voxels densely represent both empty space and surface, namely free space, leading to a lot of wasted graphics memory.

Although tree-based hierarchical data structures [8–10] can effectively subdivide space and avoid graphics memory waste, they are not effectively parallelizable when the computational complexity increases. Point-based methods [11–17], on the other hand, do not require spatial data structures, but the reconstruction quality has trouble matching the results achieved by volume-based reconstruction methods. To address these issues, a new data structure has been proposed by Nießner et al. [7] and Kähler et al. [18], which subdivides space into a group of sparse sub-blocks and uses hash functions to access them efficiently. Building on this data structure, ref. [19] combines the features of tree structures to provide surface representations at different resolutions, which effectively solves the problems of scene expansion and memory occupation in reconstruction.

One of the challenges of conducting real-time RGB-D-based 3D reconstruction on MAVs is their limited computing and storage resources. Previous vision-based 3D reconstruction systems on drones have achieved success only by transmitting data back to a ground station for processing [20–23], generating only sparse maps for navigation online [24–27], or only being able to reconstruct small-scale 3D maps [28–31] onboard. These systems can operate onboard but are prone to failure during fast drone movement, which is another focus of this article and also another challenge in conducting RGB-D-based real-time 3D reconstruction on drone platforms.

### 2.2. Pose Estimation of Fast Camera Motion

Although the fast-moving advantages of MAVs have undoubtedly heightened task efficiency, they have also presented substantial challenges in estimating their motion states. Fast camera movement poses two major challenges to camera pose estimation. Firstly, fast camera motion causes significant rotation, rendering the optimization of camera pose

highly nonlinear. When seeking to optimize the pose using gradient descent, it is easy for the optimization to become trapped in a local optimum. Secondly, fast camera motion can lead to serious motion blur in RGB images, especially under dark lighting conditions and when taking close-up shots. The motion blur in images makes it difficult to perform reliable RGB feature tracking, which is catastrophic for many feature-based SLAM methods [32,33] and dense RGB-D 3D reconstruction methods [3].

　　Numerous researchers have adopted various approaches from different perspectives to address the problem of fast-moving camera pose estimation. At the camera level, a straightforward method to mitigate the adverse effects of image motion blur is to increase the camera's frame rate. However, high frame rates lead to brief exposure times that lower the image's signal-to-noise ratio, especially under dim lighting conditions [34,35]. The sudden surge in data volume within a short period significantly increases the computational cost of the system, which is unfavorable for real-time algorithm execution. Saurer et al. [36] have also considered the jelly effect caused by fast-moving rolling-shutter cameras. Another study used event cameras for fast-moving pose estimation [37]. At the image level, researchers have attempted to minimize the negative effects of motion blur on the image to the maximum extent possible by performing image deblurring before feature extraction [38,39]. Unfortunately, considering the computational cost, these methods are challenging to use in real-time 3D reconstruction systems. Introducing additional information, such as fusing *Inertial Measuring Unit* (IMU) data, is also an effective method for estimating fast-moving camera pose. The IMU provides acceleration data at high frequencies and can serve as good initialization to predict inter-frame motion during gradient descent pose optimization [40]. Considering the high cost of IMUs, some researchers have sought to use low-cost IMU sensors to assist pose estimation [41]. However, the gyroscope sensors of IMUs, especially those built into commodity RGB-D cameras, are more effective in measuring directional changes than estimating translation. Many researchers have found that translation errors are too large to be used for tracking, either serving for attitude initialization [18,42,43] or for joint optimization [44].

　　Another unique approach is to utilize the *Particle Filter Optimization* (PFO) algorithm, which tracks the camera's rapid movements solely based on depth images. This is because fast camera movements may cause motion blur in RGB images but have a smaller impact on depth images. Fast camera moving often results in depth value overshoot or undershoot at the foreground and background transitions, rather than mixed pixel depth values across the entire image [45]. The basic idea of the PFO algorithm is to transform the objective function into a target *Probability Density Function* (PDF), and then to simulate the target PDF through sequential importance sampling. It is hoped that the optimal value of the objective function can be covered by the sampled particles. *Particle Swarm Optimization* (PSO) randomly generates a set of particles and drives them to move towards good local optima based on the designed system dynamic function. Ji et al. [46] use the update and optimization of particle swarm as a system dynamic to drive the movement of the particle ensemble. However, due to the high cost of continuous sampling and updating of particles, it has not been widely adopted in real-time applications. In a recent attempt, the problem of dense particle sampling and updating in standard PFO that affects the real-time performance of the system was solved by moving and updating a pre-sampled *Particle Swarm Template* (PST) instead of sequential importance sampling [4]. This method shows good real-time performance on a ground workstation, but still cannot run in real-time on airborne computing hardware platforms. We have provided a table comparing the most representative real-time RGB-D reconstruction systems in terms of four key capabilities: fast-moving tracking, sparse representation, scalable reconstruction, and onboard performing, as shown in Table 1. Our proposed method combines the sparsity of surface representation to enhance PFO, aiming to reduce system complexity, accelerate optimization convergence speed, and overcome certain limitations present in existing systems.

**Table 1.** State-of-the-art real-time dense 3D reconstruction systems based on RGB-D camera. ✓ denotes the presence of the specific capability, while ✗ denotes the absence of the specific capability.

| Systems | Fast-Moving Tracking | Sparse Representation | Scalable Reconstruction | Onboard Performing |
|---|:---:|:---:|:---:|:---:|
| KinectFusion [1] | ✗ | ✗ | ✗ | ✗ |
| Kintinuous [2] | ✗ | ✗ | ✓ | ✗ |
| Voxel Hashing [7] | ✗ | ✓ | ✓ | ✗ |
| InfiniTAM [15] | ✗ | ✓ | ✓ | ✓ |
| Hierarchical voxels [19] | ✗ | ✓ | ✓ | ✗ |
| BundleFusion [3] | ✗ | ✗ | ✓ | ✗ |
| RoseFusion [4] | ✓ | ✗ | ✗ | ✗ |

## 3. Methodology

The input for onboard 3D reconstruction consists of a sequence of RGB-D frames captured in real-time by RGB-D cameras, denoted by $\{I_c, I_d\}_{k=0:K}$ where $I_c$ and $I_d$ represent the RGB and depth images, respectively. The output is the surface reconstruction $\mathcal{S}$ of the captured scene and the 6 *Degrees of Freedom* (6DoF) camera pose trajectory $\left\{\left[\mathbf{R}^k \middle| \mathbf{t}^k\right]\right\}_{k=0:K}$, where $\mathbf{R}^k \in \mathbb{SO}_3$ and $\mathbf{t}^k \in \mathbb{R}^3$ represent the 3D rotation and translation in the global coordinate system. We employ our method, OwlFusion, within a framework of randomized optimization [4], which is the de facto method for large-scale high-quality online dense 3D reconstruction on low-computational hardware platforms. The key challenge is to estimate the 6DoF pose of each frame and to fuse the captured surface data. The randomized optimization framework allows for fast camera motion tracking in low-light conditions. To reduce the computational cost and accelerate the optimization convergence of randomized optimization on low-computational hardware, we introduce planar constraints based on sparse representation of the scene surface, which is our key contribution. Figure 2 provides a block diagram overview of our method.

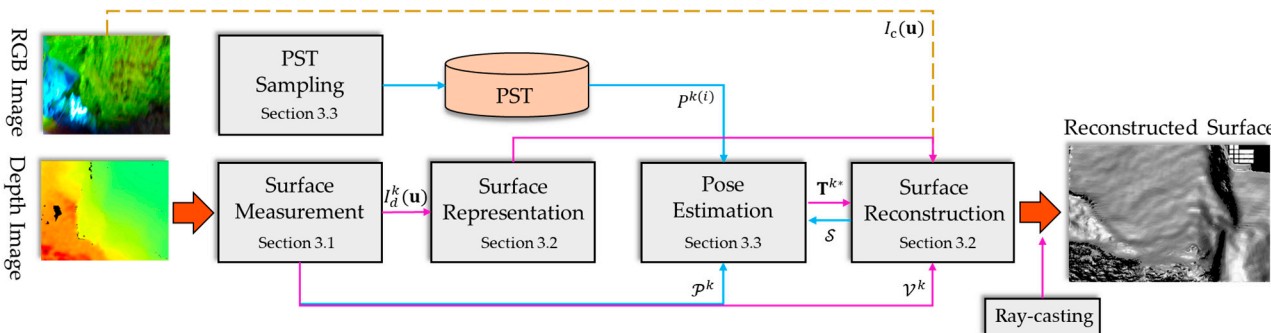

**Figure 2.** The proposed OwlFusion overview. The purple arrows indicate operations related to scene reconstruction, the blue arrows indicate operations related to pose estimation, and the yellow indicates optional operations.

Our method consists of three main parts: surface measurement, surface reconstruction, and pose estimation. In the surface measurement step, we preprocess each depth image frame input by computing the vertex map $\mathcal{V}^k$ and normal map $\mathcal{N}^k$ and generating a *Partition Normal Map* (PNM) $\mathcal{P}^k$, which we use to introduce plane constraints. In the surface reconstruction step, we adaptively allocate voxel blocks of different resolutions in GPU memory for surface representation based on the correlation between the measured depth values $I_d^k(\mathbf{u})$ and the vertex normals, achieving sparse representation of the scene. Based on the allocated voxel space and the pose $\underline{\mathbf{T}^{k*}}$ of the depth images, we continuously weight and fuse the measured depth frames into the volume to reconstruct the scene surface $\mathcal{S}$. The quality of the scene reconstruction heavily relies on the accuracy of the pose estimation. In the pose estimation step, we evaluate the fitness of random particles $P^{k(i)}$ based on the

reconstructed sparse surface $\mathcal{S}$ and PNM $\mathcal{P}^k$, which serves as a constraint to alleviate the computational burden and accelerate the optimal pose estimation speed. Finally, we use the classical ray-casting method [1] to extract the scene surface.

### 3.1. Surface Measurement

Surface Measurement is the first step in our method, which takes the raw depth image $I_d$ as input. We use $\mathbf{u} = (x, y)^T \in \mathbb{R}^2$ to represent a two-dimensional pixel on $I_d$. Given the camera intrinsic parameter matrix $\mathbf{K}$, we convert each depth measurement $I_d(\mathbf{u})$ into the three-dimensional position $v(\mathbf{u})$ of a vertex in the camera coordinate system using

$$v(\mathbf{u}) = I_d(\mathbf{u})\mathbf{K}^{-1}\left(\mathbf{u}^{\mathrm{T}}, 1\right)^{\mathrm{T}} \in \mathbb{R}^3, \tag{1}$$

which forms the vertex map $\mathcal{V}^k$ corresponding to the depth map. We then determine the normal map $\mathcal{N}^k$ at each vertex in the vertex map by computing

$$n(\mathbf{u}) = \boldsymbol{v}_{hor} \times \boldsymbol{v}_{ver}, \tag{2}$$

where

$$\boldsymbol{v}_{hor} = (v(x-1, y) - v(x+1, y)) \in \mathbb{R}^3 \tag{3}$$

and

$$\boldsymbol{v}_{ver} = (v(x, y-1) - v(x, y+1)) \in \mathbb{R}^3 \tag{4}$$

represent the direction vector of the three-dimensional points on both sides of the point $\mathbf{u}$ horizontally and the direction vector of the three-dimensional points on both sides of the point $\mathbf{u}$ vertically, respectively. We normalize the cross-product result to obtain the normal vector $n(\mathbf{u})$ at the current point. At this point, the direction of the normal vector points away from the camera center, so we flip the direction to point towards the camera center. Points located at the edge of the depth image are not used to calculate the normal vector. Here, $v(\mathbf{u})$ and $n(\mathbf{u})$ represent the elements in the vertex map $\mathcal{V}^k$ and normal map $\mathcal{N}^k$, respectively.

In OwlFusion, a PNM $\mathcal{P}^k$ is generated based on the normal map $\mathcal{N}^k$ to introduce planar constraints, as shown in Figure 3. The PNM can be considered as the result of grouping pixels in the normal map $\mathcal{N}^k$ based on their similarity in the normal direction. It clusters adjacent pixels based on their similarity in the normal direction and visualizes the clustering result as different-colored regions, each representing a plane. In this process, we use a growth method to obtain the PNM $\mathcal{P}^k$. Specifically, we choose a pixel in the normal map as the seed pixel and calculate the normal angle and Manhattan distance between it and its neighboring pixels. If both evaluation criteria are below a given threshold, the neighboring pixel is accepted as a region growth unit. We choose the normal angle threshold as $0.6°$ and the Manhattan distance threshold as the sum of the image's width plus height based on experience. After growth, if the number of pixels in a segmented region is less than 4% of the total image pixels, the region is rejected; otherwise, the segmented region is too small. In a region growth process, the seed pixel and the pixels in the segmented region will not be repeatedly calculated. Through the segmented normal map, we can quickly drive the particle swarm to move to the vicinity of the optimal pose, thereby accelerating the optimization convergence. Compared to directly extracting the scene plane on the depth map, the normal map-based region growth method is more robust to depth map measurement noise and faster in parallel computing on graphics processing units.

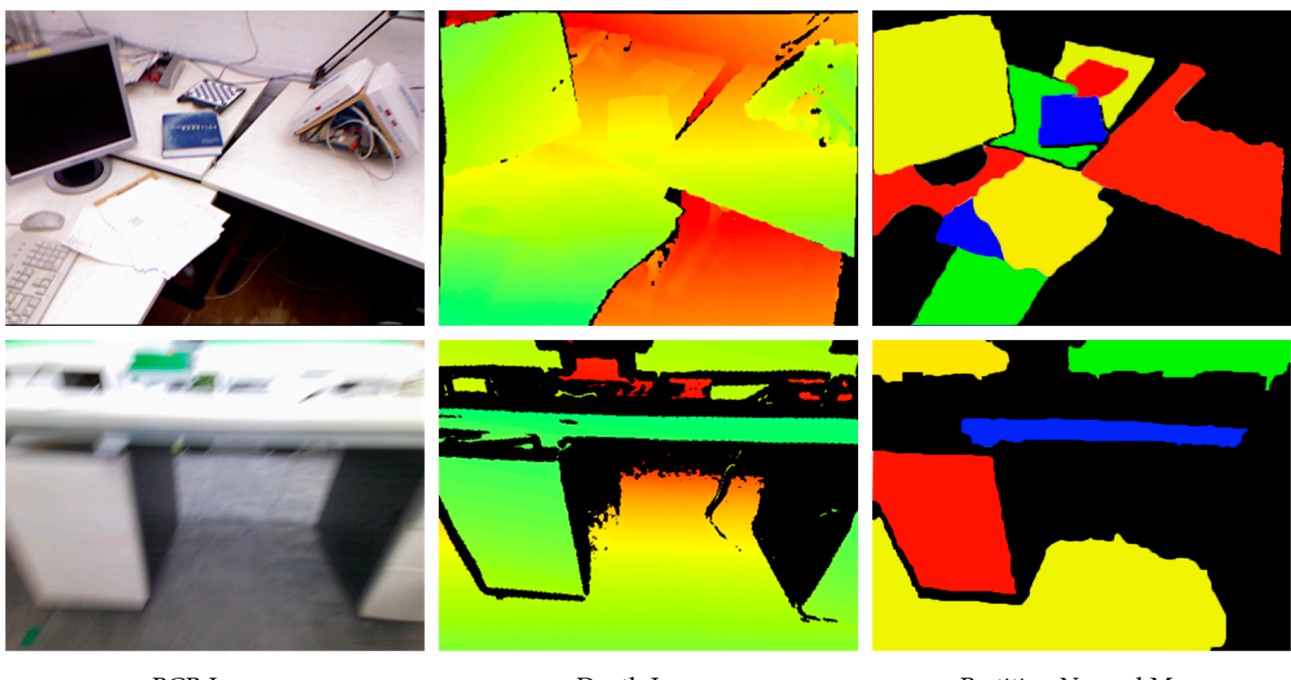

| RGB Image | Depth Image | Partition Normal Map |

**Figure 3.** The PNMs generated for two different viewpoints. The top row displays the frames captured with slow camera motion, while the bottom row displays the frames captured with fast camera motion, where noticeable motion blur can be observed in the RGB images.

### 3.2. Surface Reconstruction

The design requirements for adaptive resolution fast surface reconstruction include efficient hierarchical voxel allocation and noise-robust depth data fusion methods. To achieve this, we build upon previous work that uses a fixed number of L-level resolution layers to store surface voxels [19] and implement effective access to each level's voxels using a hash table [7,18,19]. However, instead of uniform voxel allocation, we selectively allocate voxel blocks to pixels in different regions using a PNM as a mask. For pixels in planar regions, we allocate voxel memory at the coarsest resolution level, while for non-planar regions, we skip the coarsest level and directly allocate voxel memory at the next coarsest level. On the coarsest level hash table entry, we implicitly refer to the position of the sub-blocks in finer resolution levels by a special marker.

To reconstruct the surface, we first dynamically allocate memory for voxels within the camera field of view and then use voxel splitting and merging to achieve a hierarchical representation of the surface. When a new depth frame $k$ is input into our system, we construct a ray $\mathcal{R}$ for each depth measurement $I_d(\mathbf{u})$ with respect to the camera center, where

$$\mathcal{R} = \mathbf{T}_{g,k-1}\mathbf{K}^{-1}\left(\mathbf{u}^{\mathrm{T}}, 1\right)^{\mathrm{T}}. \tag{5}$$

Here, $\mathbf{T}_{g,k-1} = \begin{bmatrix} \mathbf{R}_{g,k-1} & \mathbf{t}_{g,k-1} \\ \mathbf{0}^T & 1 \end{bmatrix} \in \mathbb{SE}_3$ is the transformation from frame $k$ in the camera coordinate system to the global frame $g$, where $\mathbf{R}_{g,k-1} \in \mathbb{SO}_3$ is the rotation transformation matrix and $\mathbf{t}_{g,k-1} \in \mathbb{R}^3$ is the translation transformation vector of the previous frame. Given the truncation range $\mu$ of the TSDF [6], we create a line segment along the direction of ray $\mathcal{R}$ within the truncation band of $I_d(\mathbf{u}) - \mu$ to $I_d(\mathbf{u}) + \mu$. For voxels that intersect with this line segment, we create a corresponding entry in the hash table and allocate memory for the unallocated voxel blocks on the GPU [7,18,19]. After voxel memory allocation is complete, we compute the roughness $r(b)$ of the surface on

which the voxels reside, which serves as the criterion for voxel splitting and merging. We define the roughness of the surface using the correlation of normals, calculated as

$$r(b) = \frac{1}{m-1} \sum_{i=1}^{m} \left( \Delta F_m(i) - \overline{\Delta F_m} \right)^2,$$

(6)

where $\Delta F_m$ represents the part of the voxel block $b$ that stores TSDF and $m$ is the number of TSDF values. $\Delta$ is the gradient operator, $\Delta F_m(i)$ represents the normal at voxel $i$, and $\overline{\Delta F_m}$ represents the average normal of $m$ voxels. It should be noted that we use the gradient of TSDF values to calculate voxel normals, rather than directly using normals from the normal map derived from depth measurements. This is because the normal map from depth measurements contains sensor noise, and the TSDF values in voxels are the results of multiple weighted fusion observations with high credibility. When the roughness is greater than the segmentation threshold $t_f$, the current level voxel is segmented. When the roughness is less than the merging threshold $t_h$, the current level voxel is merged, and the resolution level of each voxel is recorded.

After allocating all voxel blocks within the truncation region, the current depth frame is fused with the reconstructed visible surface voxels. To efficiently perform the fusion of TSDF, we first access all entries in the hash table before fusing the depth frames, selecting the hash entries that point to visible voxel blocks within the camera view frustum [7], thereby avoiding empty voxel blocks in the hash table. These hash entries are then processed in parallel to update the TSDF values. The global fusion of all depth maps in the volume is formed as the weighted average of all individually calculated TSDFs from each depth frame, which can be viewed as denoising the global TSDF from multiple noisy TSDF measurements. We adopt the TSDF fusion framework of [18], but redefine the weight $W_k$ of the TSDF. Considering the effect of sensor noise, we define

$$W_k(v(\mathbf{u})) = \exp\left( \frac{-\gamma^2 \cos\theta}{2\delta^2 I_d(\mathbf{u})} \right),$$

(7)

where $\gamma$ is the normalized radial distance of the current depth measurement $I_d(\mathbf{u})$ from the camera center, and $\delta = 0.6$ is derived empirically. We define an imaging validity factor and a scan validity factor, respectively,

$$\alpha = -\gamma^2 / 2\delta^2$$

(8)

$$\beta = \cos\theta / I_d(\mathbf{u}),$$

(9)

where $\theta$ is the angle between the ray direction of the depth pixel $\mathbf{u}$ and the normal measurement of the corresponding surface point $v(\mathbf{u})$ in the local frame. If the depth measurement is within the valid distance range and the scan angle of the visible surface points in the depth map is $0°$, the validity of the point is maximum $\beta = 1$, which decreases as the scan distance exceeds the valid range or deviates from $0°$. As the reconstructed surface may extend beyond or revisit the camera view frustum during system running, a bidirectional GPU–Host data stream scheme [7,18] is used to store the reconstructed voxels that exceed the current camera view frustum in the host, allowing the system to fully utilize the limited GPU memory and performance and enable unlimited reconstruction. When the camera returns to a previously reconstructed position, the voxels stored in the host in that region are streamed back to the GPU for fusion and reuse.

### 3.3. Pose Estimation

Accurate image pose estimation is crucial for surface reconstruction using depth images. However, traditional methods [3,14,33,47–51] for pose estimation on MAV platforms can fail to track the camera due to fast camera motion, resulting in incorrect pose estimation results. To address this challenge, our work uses a particle swarm template random optimization [4]. Unlike previous work, we introduce planar constraints with the help of PNM (Section 3.1) based on hierarchical sparse surface representation (Section 3.2). Our method performs *Candidate Particle Set* (CPS) filtering firstly before particle fitness evaluation, and subsequent iterations of particle optimization select particles from the CPS. Compared to the *Advantage Particle Set* (APS) defined in [4], the CPS is a much smaller particle swarm set that is strictly constrained, which helps to reduce computational cost during random optimization and accelerate the convergence of pose optimization iterations.

To reflect the alignment between the current and the previous frame accurately, we need to determine the overlapping area between the planar regions of the two frames. To identify the set of overlapping pixels $O^k$ between the PNMs $\mathcal{P}^k$ and $\mathcal{P}^{k-1}$, we adopt an unproject-and-reproject approach:

$$O^k = \left\{ (i,j) \left| \mathbf{T}^{k-1} \left( \mathbf{T}^k \right)^{-1} [(i,j)] \in \mathcal{P}^{k-1} with (i,j) \in \mathcal{P}^k \right. \right\}, \tag{10}$$

where $\mathbf{T}^k$ is the projection matrix of frame $k$ under the camera pose $P^k$. Based on $O^k$, CPS is calculated by projecting the overlapping pixels onto the volume and normalizing based on whether the corresponding voxel is in the coarsest voxel level:

$$\Omega_c^k = \left\{ P^{k(i)} \in \Omega \left| \frac{N_l}{|O^k|} > 0.96 \right. \right\}, \tag{11}$$

where $P^{k(i)}$ represents any pose particle in the PTM $\Omega$ and $N_l$ is the number of overlapping pixels projected onto the coarsest voxel level. Due to the uncertainty of edge pixels in the PNM plane segmentation, we relax the percentage of $N_l$ relative to the total overlapping pixels to 96% based on the empirical values obtained in the experiment to ensure coverage of the optimal solution. Meanwhile, we use the same method to identify the set of overlapping pixels $Q^k$ between the depth frames $I_d^k$ and $I_d^{k-1}$:

$$Q^k = \left\{ (i,j) \left| \mathbf{T}^{k-1} \left( \mathbf{T}^k \right)^{-1} [(i,j)] \in I_d^{k-1} with (i,j) \in I_d^k \right. \right\}. \tag{12}$$

In each iteration $t$ during the optimization of $P^k$, $Q_t^k$ is used as the valid pixel set to evaluate the particle fitness:

$$\rho \left( P_t^{k(i)} \right) = \exp \left( - \frac{\sum_{(i,j) \in Q_t^k} \psi \left( \mathbf{R}_t^k \mathbf{x}_{ij} + \mathbf{t}_t^k \right)}{|Q_t^k|} \right), \tag{13}$$

where $\mathbf{R}_t^k$ and $\mathbf{t}_t^k$ represent the rotation and translation of the pose particle $P_t^{k(i)} \in \Omega_c^k$, and $\psi(\cdot)$ represents the so far constructed TSDF. Note that the inter-frame overlap is deliberately maintained at a non-negligible level to avoid any potential over-evaluation of poses with minimal overlap. Our PST scaling scheme ensures that the sampled transformation, relative to the pose of the previous step, remains within a controlled range of 10 cm in translation and $10°$ in rotation. This careful constraint ensures that the evaluations remain within appropriate bounds and accurately reflect the relevant transformations. In each iteration, we use the method in [4] to scale and move PST until we find the optimal pose $\mathbf{T}^{k*}$, i.e.,

$$\mathbf{T}^{k*} = \left( P_t^{k(i)} \left| \max \left( \rho \left( P_t^{k(i)} \right) \right) \right. \right). \tag{14}$$

The experiments will demonstrate that our method requires significantly fewer iterations to find the optimal solution compared to the method in [4] while also reducing the time and resource consumption for pose estimation.

## 4. Experiments

In this section, we first introduce the hardware information and computational performance of our platform, as well as the settings of experimental parameters. Then, we describe the dataset used in our experiments. After that, we evaluate the effectiveness of planar constraints in improving the efficiency of random optimization for pose estimation and the importance of imaging validity factor and scan validity factor weighting for surface reconstruction, as well as the memory usage and processing efficiency of our entire system. Finally, we compare our onboard real-time 3D reconstruction method with state-of-the-art methods through qualitative and quantitative experiments.

### 4.1. Performance and Parameters

We conducted all experiments on an embedded computing device, the Nvidia Xavier NX. This device is equipped with a six-core ARM Cortex-A57 CPU and a dual-core NVIDIA Denver 2.0 CPU, along with 8 GB of LPDDR4x RAM and an NVIDIA Volta GPU with 512 CUDA cores, providing a processing capacity of 1.3 TFLOPS. Furthermore, it is equipped with 16 GB of high-speed HBM2 memory, which offers fast data transfer and processing capabilities. We set the basic voxel size to $b_0 = 2$ mm, which provides a very high level of detail. In our hierarchical representation, we used three levels of resolution, resulting in a coarsest voxel size of $b_2 = 8$ mm. The truncation distance for the TSDF was set to $\mu = 24$ mm. To increase the credibility of our experiments, we did not add any additional acceleration processing to the proposed method, kept the consistency of the experimental data input, and left all other parameters at their default values unless otherwise specified.

### 4.2. Benchmark

We defined a fast camera motion as having a linear velocity greater than 1 m/s or an angular velocity greater than 2 rad/s. We found two publicly available datasets, FastCaMo [4] and FMDataset [52], that satisfy this definition. FastCaMo is the first RGB-D sequence dataset specifically designed for fast camera motion, comprising synthetic (FastCaMo-Synth) and real captured (FastCaMo-Real) parts, as well as ground truth trajectories and reconstructions for evaluating system performance. FMDataset not only provides color and depth images captured by a depth camera but also IMU information of the camera. We also evaluated our method on traditional RGB-D datasets, including TUM RGB-D [53], ICL-NUIM [54], and ETH3D [55]. The motion speed of the former two is lower than our defined fast camera motion (usually less than 1 m/s), while in ETH3D, we focus on three sequences with the prefix "camera shake", all of which have angular velocities above 2.5 rad/s. In addition, we captured four RGB-D sequences in our flight experiments, namely, "corridor1_slow", "corridor1_fast", "corridor2", and "courtyard". Table 2 provides speed information for all datasets being tested. Note that the speed of the publicly available datasets comes from ground truth camera trajectories obtained by a visual motion capture system, while the speed of FastCaMo-Real, FMDataset, and our captured real-world RGB-D sequences comes from successfully tracked camera trajectories, as it is difficult for a visual motion capture system to track such fast camera motion.

### 4.3. Evaluation

In this section, we conducted comprehensive ablation experiments to evaluate key design aspects of our system. This included assessing the effectiveness and efficiency of the planar constraint in the random optimization process, evaluating the benefits of a hierarchical sparse surface representation for scalable reconstruction, and analyzing the weighted depth fusion scheme incorporating imaging and scan validity. Additionally, we

assessed the efficiency of the overall system. These experiments provide valuable insights into the performance and significance of each component in our proposed method.

**Table 2.** Statistics on camera moving speed (average linear velocity $\overline{v}$, maximum linear velocity $v_{max}$, average angular velocity $\overline{\omega}$, and maximum angular velocity $\omega_{max}$) for different benchmark datasets.

| Sequence | $\overline{v}$ (m/s) | $v_{max}$(m/s) | $\overline{\omega}$ (rad/s) | $\omega_{max}$(rad/s) |
|---|---|---|---|---|
| TUM_fr1/desk | 0.41 | 0.66 | 0.41 | 0.94 |
| TUM_fr1/room | 0.33 | 0.76 | 0.52 | 0.85 |
| TUM_fr3/office | 0.25 | 0.36 | 0.18 | 0.35 |
| ICL_lr_kt0 | 0.13 | 0.27 | 0.16 | 0.33 |
| ICL_lr_kt1 | 0.05 | 0.09 | 0.10 | 0.40 |
| ICL_lr_kt2 | 0.28 | 0.40 | 0.23 | 0.46 |
| ICL_lr_kt3 | 0.27 | 0.38 | 0.12 | 0.41 |
| ETH3D_camera_shake1 | 0.46 | 0.64 | 1.88 | 2.65 |
| ETH3D_camera_shake2 | 0.33 | 0.48 | 1.90 | 3.27 |
| ETH3D_camera_shake3 | 0.37 | 0.51 | 2.16 | 3.43 |
| FastCaMo_real/lab | 0.98 | 3.62 | 0.91 | 5.20 |
| FastCaMo_real/apartment1 | 1.05 | 4.22 | 1.08 | 5.73 |
| FastCaMo_real/apartment2 | 1.71 | 3.73 | 1.38 | 4.21 |
| FastCaMo_synth/apartment1 | 1.53 | 3.88 | 0.92 | 2.08 |
| FastCaMo_synth/hotel | 1.66 | 3.94 | 1.13 | 2.23 |
| FMDataset_dorm1_fast1 | 0.52 | 0.92 | 1.24 | 2.59 |
| FMDataset_dorm2_fast | 0.75 | 1.60 | 1.23 | 2.16 |
| FMDataset_hotel_fast1 | 0.75 | 1.26 | 1.29 | 2.34 |
| FMDataset_livingroom_fast | 0.53 | 1.77 | 0.85 | 2.41 |
| FMDataset_rent2_fast | 0.83 | 1.54 | 1.31 | 2.27 |
| Ours_corridor1_slow | 0.40 | 0.73 | 0.55 | 0.89 |
| Ours_corridor1_fast | 0.92 | 1.44 | 1.31 | 3.03 |
| Ours_corridor2 | 0.87 | 1.95 | 1.42 | 2.89 |
| Ours_courtyard | 1.01 | 2.30 | 1.09 | 3.37 |

### 4.3.1. Random Optimization with Planar Constraint

To evaluate the effectiveness and accuracy of the proposed planar-constrained stochastic optimization method for tracking fast camera movements, we compared stochastic optimization methods with and without planar constraints [4]. The method without planar constraints involves sampling particles directly around the best pose from the previous frame for particle fitness evaluation and iteratively computing APS to obtain the optimal solution. The iteration counts of the two methods for processing the same dataset were statistically analyzed on different test datasets, as shown in Figure 4.

Compared to the original approach, the introduction of planar constraints significantly reduces the number of iterations required to optimize each frame. This greatly improves the efficiency of the optimization process. Due to the CPS selection mechanism, the range of randomly selected particles is significantly narrowed, resulting in a set of particles that are closer to the optimal pose in the initial iteration and to some extent avoid getting trapped in local optima. Our implementation ensures that the random optimization process converges quickly with a minimum number of iterations. When we set the termination condition to "APS is empty for two consecutive iteration steps" and "the change of the optimal pose $T^{k*}$ in 6DoF is less than $1 \times 10^{-6}$ for two consecutive iteration steps", our optimization algorithm typically converges in less than two iterations for slow motion (<1 m/s) and in less than five iterations for fast motion ($\geq$1 m/s).

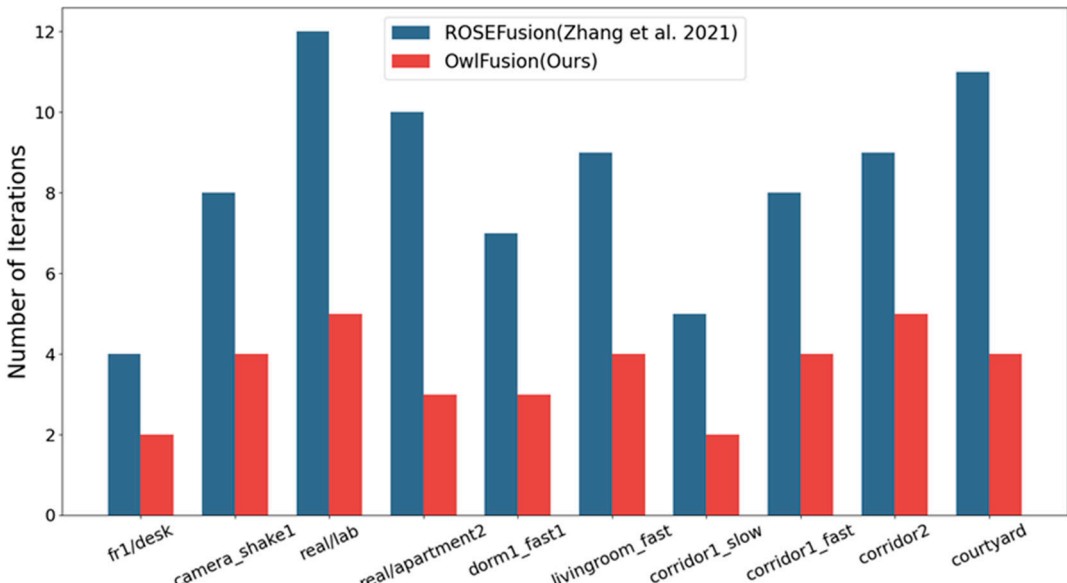

**Figure 4.** Comparison of average iteration counts per frame between the two random optimization methods with and without the proposed planar constraint [4].

We further validated the efficiency of our implementation by calculating the average processing time per frame *t*, as shown in Table 3. Additionally, the table presents the *Absolute Trajectory Error* (ATE) between our method and the comparison methods against the ground truth trajectory, as well as the *Mean Distance* (MD) between the reconstructed and ground truth models. The *Root Mean Square Error* (RMSE) of the ATE for slow motion sequences was calculated using the tool provided by Sturm et al. [53]. Due to the difficulty of capturing such fast camera motion with a visual motion capture system, there is no ground truth trajectory available in the public datasets for our captured sequences. Hence, we indirectly assessed the accuracy of pose estimation by utilizing high-precision 3D dense laser-scanned reconstruction models provided by Zhang et al. [4] as ground truth. The accuracy evaluation was conducted by calculating the MD between the reconstructed model and the laser-scanned model, employing the open-source software CloudCompare version 2.11.3.

**Table 3.** Comparing the efficiency and accuracy of pose estimation on whether to introduce planar constraint. The time and accuracy best results for each sequence are highlighted in blue color.

| Sequence | *t* (ms) | | ATE (cm) | | MD (cm) | |
|---|---|---|---|---|---|---|
| | **RoseFusion** | **OwlFusion** | **RoseFusion** | **OwlFusion** | **RoseFusion** | **OwlFusion** |
| fr1/desk | 218.38 | 23.35 | 2.48 | 1.93 | — | — |
| fr1/room | 219.04 | 24.20 | 4.86 | 4.32 | — | — |
| fr3/ office | 209.98 | 23.81 | 2.51 | 2.63 | — | — |
| lr_kt0 | 214.63 | 24.73 | 0.83 | 0.77 | — | — |
| lr_kt1 | 212.69 | 24.55 | 0.71 | 0.80 | — | — |
| camera_shake1 | 224.24 | 27.09 | 0.62 | 0.93 | — | — |
| camera_shake2 | 227.84 | 26.64 | 1.35 | 1.07 | — | — |
| camera_shake3 | 232.18 | 29.39 | 4.67 | 4.54 | — | — |
| synth/apartment1 | 228.93 | 29.60 | 1.10 | 1.32 | 4.52 | 4.37 |
| synth/hotel | 230.62 | 30.08 | 1.52 | 1.33 | 5.25 | 5.54 |
| real/lab | 230.24 | 30.27 | — | — | 4.86 | 4.50 |
| real/apartment1 | 230.75 | 30.51 | — | — | 4.88 | 5.45 |
| real/apartment2 | 228.69 | 24.20 | — | — | 4.23 | 5.01 |

Compared to the original random optimization method for pose estimation, the method with planar constraints improves the efficiency of pose estimation by about 8 times while maintaining the same level of accuracy. The time it takes to estimate the pose increases as the camera's velocity increases since more optimization iteration steps are required to ensure accurate and stable tracking of fast-moving cameras. However, our pose estimation method and the comparison method do not have the same iteration time consumption, as this depends on the chosen strategy for particle sets and fitness evaluation. While we adopted the same approach to assess the particle fitness as the comparison method, we pre-selected a particle set with a smaller CPS of planar constraints to significantly reduce computational costs during the random optimization process and accelerate pose optimization convergence. The improvement in pose estimation efficiency is due to our estimation algorithm, as well as the GPU parallel computing and the hierarchical sparse data structure we used. It is worth noting that in our tests, we only used sequence frames that could be reconstructed with the same range as the comparison method. The reason for this is that the method lacked scene scalability, which we addressed in our implementation.

4.3.2. Scalability and Quality of Scene Reconstruction

To evaluate the scalability of our proposed method, we compared its ability to reconstruct long RGB-D sequences with regular volume reconstruction approaches. Figure 5 presents and compares the reconstruction outcomes of both methods on real-world scenes captured by our MAV. The sequences' ground length exceeded 30 m, with corridor 1 and courtyard exceeding 50 m. Our proposed method produced complete reconstructions of the entire sequences in all three scenes, while the comparison method only provided partial reconstructions. This limitation stemmed from the regular volume surface reconstruction approach's need to predefine the reconstruction range before surface reconstruction, as well as the limited GPU memory available in airborne computing devices, which resulted in a much smaller maximum reconstruction range than that of base stations. However, our approach did not require such presetting of the reconstruction range and enabled timely data exchange between the GPU and host via the adopted bi-directional GPU–Host data exchange approach, thus freeing up space for subsequent surface reconstruction. Additionally, the random optimization pose estimation in regular volume reconstruction experienced a decline in precision when the camera approached the edge of the predefined reconstruction boundaries, leading to reconstruction misalignment, as demonstrated by the red box in Figure 5.

To validate whether the imaging and scan validity weighting introduced is useful in reducing sensor noise in deep fusion, we compared the reconstruction details of two methods, one with weighting and the other without. Figure 6 depicts the partial reconstruction details of two publicly available datasets. Our approach measures the observation validity in the deep fusion, thus obtaining high-quality denoising modeling from defective data. In an extreme case, as shown in Figure 7, there is a defect in scanning the left part of the teddy bear, causing significant noise on the reconstructed surface (as indicated by the red circle). On the contrary, our weighted fusion method demonstrated better modeling performance.

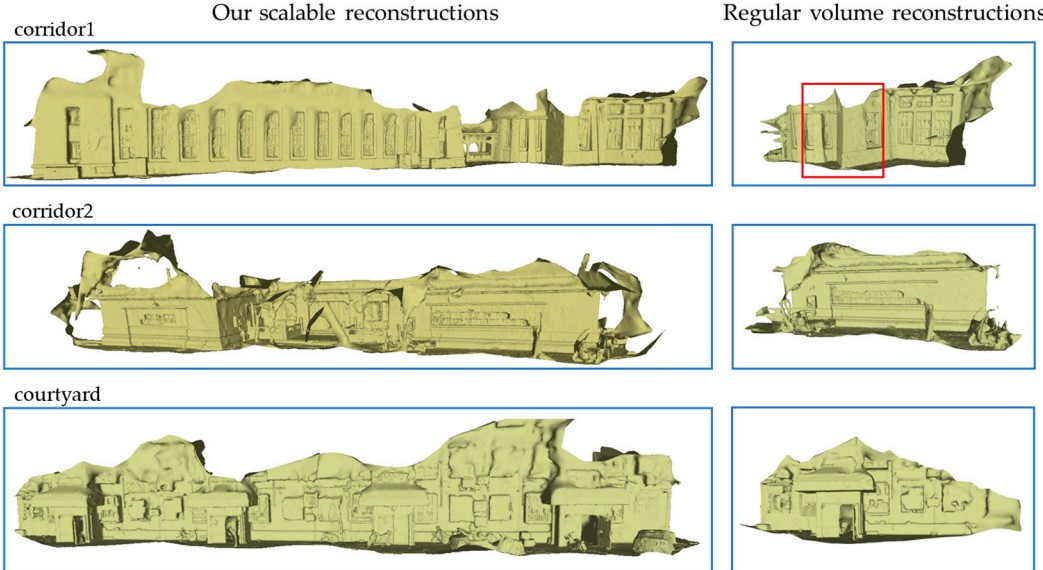

**Figure 5.** Gallery of 3D reconstruction results for the three real captured sequences by our MAV. For each sequence, we compare the scalability of our reconstruction method and regular volume construction method (other modules are consistent with our system).

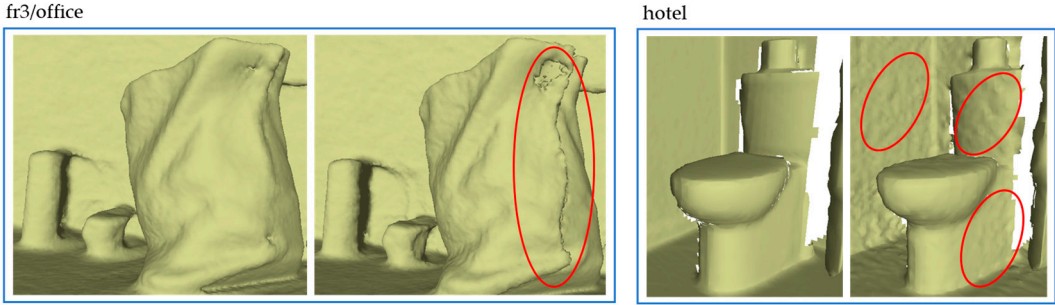

**Figure 6.** Gallery of 3D reconstruction details for the two publicly available datasets. In the comparison figure, the left side considers weighting by imaging and scan validity, while the right side does not.

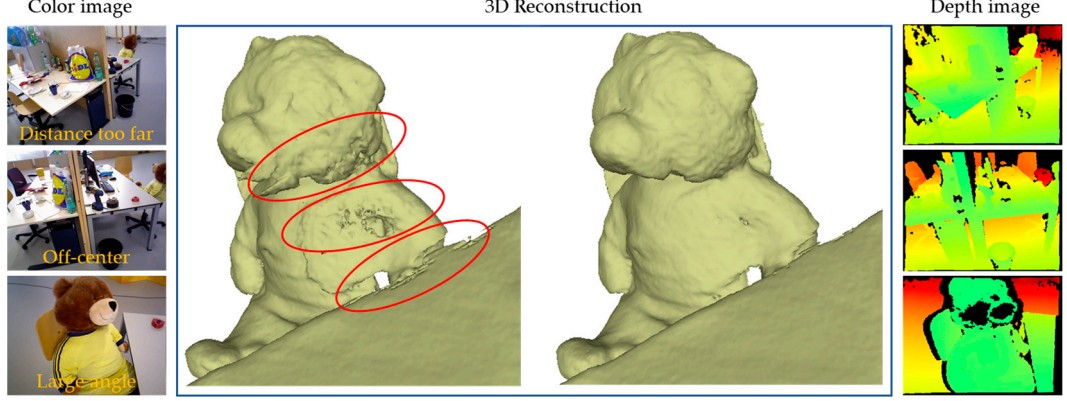

**Figure 7.** An extremely unfavorable observation: throughout the entire scanning sequence, the left side of the teddy bear's body remains at a too far distance from the camera (Distance too far), deviated from the camera center (Off-center), or with a large scanning angle (Large angle).

4.3.3. System Efficiency

Real-time performance is a critical requirement in many applications, particularly in robotics technology. Table 4 presents typical frame rates achieved by our CUDA parallel processing implementation, measured on Nvidia Xavier NX GPU. In all cases, they significantly exceed the real-time performance of the comparison method [4] on airborne consumer-grade graphics hardware. Our system operates in real-time at a rate of 28–37 Hz without enabling real-time visualization. When real-time visualization is enabled, the system's efficiency decreases, but it still runs at a minimum rate of 30 Hz. This is mainly due to the more complex interpolation scheme used in the hierarchical voxel representation during the ray-casting process [19]. The introduction of the effectiveness weighting increases the time consumption of the surface reconstruction step, which accounts for approximately 8.7% of the total time consumption compared to the comparison method's 7.5%, which can be negligible.

**Table 4.** Comparing the system running efficiency on different systems (the comparison method [4], our method with real-time visualization, and our method without real-time visualization), measured in *Frames Per Second* (FPS).

| Sequence | FPS (Hz) | | |
|---|---|---|---|
| | Comparison | Ours w/ Vis. | Ours w/o Vis. |
| fr1/desk | 3.50 | 30.19 | 37.18 |
| fr1/room | 3.49 | 29.14 | 35.88 |
| fr3/ office | 3.64 | 29.60 | 36.46 |
| lr_kt0 | 3.56 | 28.51 | 35.11 |
| lr_kt1 | 3.60 | 28.72 | 35.37 |
| camera_shake1 | 3.41 | 26.02 | 32.05 |
| camera_shake2 | 3.36 | 26.46 | 32.59 |
| camera_shake3 | 3.29 | 23.99 | 29.54 |
| synth/apartment1 | 3.34 | 23.82 | 29.34 |
| synth/hotel | 3.32 | 23.44 | 28.87 |
| real/lab | 3.32 | 23.29 | 28.68 |
| real/apartment1 | 3.32 | 23.11 | 28.46 |
| real/apartment2 | 3.35 | 30.19 | 37.18 |

*4.4. Comparison*

In this section, we compared our proposed method with the state-of-the-art approaches in the field. We discussed the similarities, differences, strengths, and limitations of various methods, providing readers with a thorough and critical assessment. This comparative analysis helped readers understand the unique contributions and advantages of our proposed method in relation to other existing techniques.

4.4.1. Quantitative Comparison

We quantitatively evaluated the performance of our method and several state-of-the-art methods on both normal and fast motion sequences. Table 5 compares our method with the state-of-the-art online ORB-SLAM2 [33], ElasticFusion [56], InfiniTAM [18], BundleFusion [3], and BAD-SLAM [55] on two sequences each from the slow-motion dataset, ICL-NUIM, and the fast-motion dataset, FastCaMo_synth, in terms of ATE. Our method achieved camera tracking accuracy comparable to the best-performing BundleFusion on the slow-motion sequences, which involves global pose optimization through *Bundle Adjustment* (BA), while our method does not involve any global pose optimization. As a result, our method slightly lags behind BundleFusion in terms of accuracy. Our method's advantage is best demonstrated in the fast-motion sequences, in which the average speed of camera motion is over 10 times faster than that of ICL-NUIM. Other comparison methods failed to achieve successful reconstruction with these fast-motion datasets, whereas our method still achieved tracking accuracy with an ATE of less than 1.5 cm on all three sequences.

This discrepancy arises due to the fact that the other methods rely on feature points or other photometric information for localization. However, in the case of fast motion, the resulting blurry RGB images pose challenges in providing relevant information for accurate localization.

**Table 5.** Comparing the ATE RMSE (cm) of camera tracking on the four RGB-D sequences from two datasets, ICL-NUIM and FastCaMo_synth. The best and the second-best results for each sequence are highlighted in blue and orange colors, respectively.

| Method | lr_kt0 | lr_kt1 | syn./apartment1 | syn./hotel |
|---|---|---|---|---|
| ORB-SLAM2 | 1.11 | 0.46 | — | — |
| ElasticFusion | 1.06 | 0.82 | 41.09 | 43.64 |
| InfiniTAM | 0.89 | 0.67 | 10.38 | — |
| BundleFusion | 0.61 | 0.53 | 4.70 | 65.33 |
| BAD-SLAM | 1.73 | 1.09 | — | — |
| OwlFusion | 0.83 | 0.72 | 1.08 | 1.47 |

We compared the reconstruction performance of two synthetic sequences and three real capture sequences from the FastCaMo dataset in Table 6. The ground truth reconstruction of real capture sequences was obtained from high-precision *Light Detection and Ranging* (LiDAR) scans, and synthetic sequences also had ground truth surfaces of the same type. As a result, we evaluated the completeness, accuracy, and real-time performance of the reconstructed surfaces with regard to the ground truth surfaces. Since the reconstruction accuracy measures only the MD of overlap (inlier) areas between the reconstructed surfaces and ground truth surfaces, we set the inlier threshold to 15 cm in Table 6. When the threshold was set to 5 cm, the average error was between 1 and 3 cm, and the completeness decreased by about 10%. Reconstruction quality is best reflected in completeness, and in this regard, our method is consistently superior to the two comparison methods, as shown in the visual results of the reconstruction presented in Figure 8. This is attributed to the fact that, on one hand, the comparative methods struggle to track fast motion accurately, and on the other hand, their re-localization modules struggle to function effectively solely based on blurry RGB images after tracking failure. Real-time performance is a focus of interest in the robotics community, and our system achieves an efficiency equivalent to that of InfiniTAM, the state-of-the-art high frame rate 3D reconstruction system, on mobile devices. Although the FPS is slightly lower than that of InfiniTAM, our method is significantly superior to it in terms of reconstruction completeness and accuracy. In contrast, BundleFusion requires more processing time to handle input frames on resource-limited mobile devices and even encounters program running failures due to the high computational overhead of running BundleFusion.

**Table 6.** Comparing the reconstruction completeness (Compl. %), accuracy (Acc. cm), and running efficiency (FPS Hz) on the five RGB-D sequences from the FastCaMo_synth and FastCaMo_real datasets. The best results for each sequence are highlighted in blue color.

| Sequence | InfiniTAM | | | BundleFusion | | | OwlFusion | | |
|---|---|---|---|---|---|---|---|---|---|
| | Compl. | Acc. | FPS | Compl. | Acc. | FPS | Compl. | Acc. | FPS |
| syn./apartment1 | 21.74 | 7.32 | 31.87 | 39.82 | 5.48 | 0.67 | 93.65 | 4.37 | 29.34 |
| syn./hotel | 33.13 | 6.98 | 28.33 | 47.64 | 4.90 | 0.42 | 94.57 | 5.54 | 28.87 |
| real/lab | 11.21 | 9.24 | 30.75 | 16.88 | 5.42 | — | 92.81 | 4.50 | 28.68 |
| real/apartment1 | 9.83 | 8.73 | 29.37 | 34.23 | 6.39 | — | 87.23 | 5.45 | 28.46 |
| real/apartment2 | 15.07 | 8.68 | 32.92 | 25.17 | 5.23 | — | 89.65 | 5.01 | 37.18 |

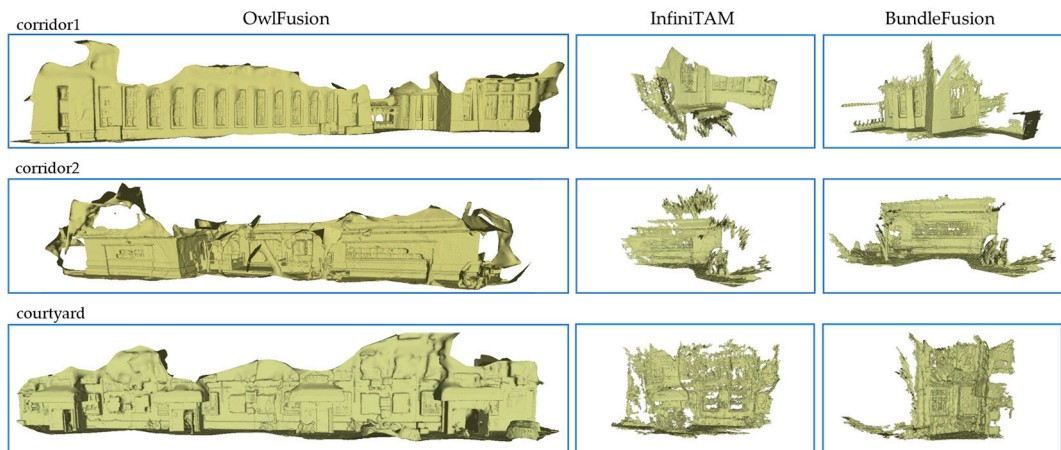

**Figure 8.** Gallery of 3D reconstruction results for the three real captured sequences by our MAV.

### 4.4.2. Qualitative Comparison

In the qualitative comparative experiment, we present the visual results of 3D reconstruction for the fast-motion dataset. Figure 8 shows the reconstruction of three real capture sequences by our MAV. For each sequence, we display the reconstruction results of OwlFusion (left), InfiniTAM (middle), and BundleFusion (right). It is evident that our reconstruction is more complete and the surface quality is acceptable compared to the comparison methods. Although BundleFusion failed to succeed on these fast-motion sequences, the few frames with successful tracking correspond to a well-reconstructed surface.

Due to the almost complete lack of loops in our dataset, which is a problem that robots must face directly in their actual applications, and our desire not to complete too much redundant work, even though both comparative methods have re-localization modules, it is difficult to recover reconstruction after losing tracking. Additionally, the aggressive camera motion results in severe image motion blur, making the loop closure of BundleFusion, which depends on color information, not work well. Although InfiniTAM successfully detects loops using depth images, it still cannot correct the significant accumulation error caused by fast camera motion. In contrast to these frame-to-frame pose estimation methods, our motion tracking uses a frame-to-model method, which has higher robustness and accuracy [2,56]. This makes our system much less prone to drift in fast motion at the same distance than some frame-to-frame methods, as shown in Tables 5 and 6. However, the accuracy of the frame-to-model pose estimation method depends on an accurate model, which forces us to pay more attention to the surface quality of reconstruction. Despite this, significant drift still occurs in the reconstruction of extremely long-distance scenes.

## 5. Limitations

Although our method has achieved excellent scalable real-time 3D reconstruction performance on low-compute devices, it still has some limitations. Firstly, the efficiency of our system just meets the "real-time" requirement, i.e., a frame rate of 30 Hz for image processing, which makes it impossible to simultaneously perform scene reconstruction and path planning algorithms onboard devices. Additionally, our method may suffer from pose estimation errors in scenes with degraded geometric features, such as graffiti walls. Joint estimation of geometric and photometric information may be a possible solution. Furthermore, within this paper, our focus has been primarily on the geometric reconstruction of scenes. However, it is crucial to acknowledge that rich texture information plays a pivotal role in achieving high-fidelity 3D reconstruction. Unfortunately, blurry RGB images pose a challenge in providing high-quality texture mapping. Finally, our system lacks the capability of global model optimization, which limits the application of high-quality ultra-long distance scene reconstruction. We believe that these limitations are worth exploring in future research.

## 6. Conclusions

In this work, we proposed a depth-only onboard scalable real-time 3D reconstruction method on MAVs. Our approach leverages two main design choices to achieve satisfactory results. Firstly, we introduced planar constraints in the random particle selection process by computing partition normal maps, reducing the computational cost of the random optimization and improving the efficiency of pose estimation. Secondly, we combined the hierarchical voxel hashing function with particle swarm optimization to further reduce the computational burden and storage cost of onboard devices, enabling real-time reconstruction of large-scale scenes. Lastly, we considered the validity of camera scanning and imaging and quantified it before incorporating it into the depth data fusion process to control the noise impact on the reconstructed surface. Future research can address the limitations of our approach, such as increasing the global model optimization capability, further improving the robustness of pose estimation while reducing the demand for computational resources, and expanding the application scenarios. Additionally, integrating other sensor data, such as stereo cameras and LiDAR, can be explored to further improve the accuracy and robustness of 3D reconstruction.

**Author Contributions:** Conceptualization, G.G.; methodology, G.G.; software, G.G. and X.W.; validation, G.G., X.W. and S.W.; formal analysis, G.G.; investigation, H.Z. and J.L.; resources, H.S.; data curation, G.G. and X.W.; writing—original draft preparation, G.G.; writing—review and editing, G.G., H.Z. and J.L.; visualization, G.G. and S.W.; supervision, H.S.; project administration, H.S. and G.G.; funding acquisition, H.S. All authors have read and agreed to the published version of the manuscript.

**Funding:** The research was funded by the Natural Science Foundation of China (NSFC) Major Program (42192580, 42192583) and the Guangxi Science and Technology Major Project (No. AA22068072).

**Data Availability Statement:** The data presented in this study are available on request from the corresponding author. The data are not publicly available due to privacy.

**Conflicts of Interest:** The authors declare no conflict of interest. The funders had no role in the design of the study; in the collection, analyses, or interpretation of data; in the writing of the manuscript, or in the decision to publish the results.

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
