# Peer review of "OwlFusion: Depth-Only Onboard Real-Time 3D Reconstruction of Scalable Scenes for Fast-Moving MAV"

_drones, doi:10.3390/drones7060358_

Round 1

Reviewer 1 Report

Comments and Suggestions for Authors

This manuscript proposes an end-to-end solution for  dense RGB-D reconstruction of scalable scenes for fast-moving MAVs. Unfortunately, I have not conducted in-depth research in this field, therefore I cannot give too many opinions of academic significance. There are two problems that can be improved in this manuscript.

1. Line 79: “With advancements in GPU architecture and GPGPU algorithms”, “GPGPU” maybe a spelling mistake?

2.The font in figure 2 had better be consistent with the legend.

Comments on the Quality of English Language

Minor editing of English language required.

Reviewer 2 Report

Comments and Suggestions for Authors

1)   Adding a table that highlights the shortcomings of other methods and research would enhance the Related Work section.

2)   An insightful discussion of the results involves going beyond a surface-level interpretation of the data and delving deeper into what the findings mean is missing. This may involve explaining how such methodology and research impact the current knowledge and state-of-the-art and the implications. This is critical and very important. 

3)   Adding a table for accornamens and abbreviations is necessary.

4)   Do not leave sections without text (e.g., 4.3. Evaluation, 4.4. Comparison). Instead, try to explain what this section is about.  

Comments on the Quality of English Language

Quality of English language is good, make sure to review and revise the paper again for grammatical errors and appropriate use of punctuation.

Reviewer 3 Report

Comments and Suggestions for Authors

This paper presents an approach for real-time 3D reconstruction using depth-only data on fast-moving drones. While the manuscript is well-written, some minor revisions are necessary.

  1. The manuscript highlights the challenge of motion blur in RGB images used for surface reconstruction. However, it does not describe how to process RGB images to eliminate or compensate for motion blur. This is crucial as blurred images cannot provide high-quality texture for the reconstructed mesh.
  2. The method relies on PNM for pose estimation, which may limit its usage scenarios. The authors should clarify what happens if there are not enough plane objects and if the method has been tested in more difficult scenarios such as low overlap or fewer textures. While the Candidate Particle Set filtering method has improved efficiency, it may be negatively affected in low texture environments, please explain.
  3. The authors define "fast" camera motion as greater than 1 m/s, which is an absolute value. However, motion blur is usually counted in pixels, and the scale is critical. The authors should provide more detailed explanations of "fast" camera motion and link it to motion blur or maybe other factors related to IMU.
  4. The authors should clearly state if there is a maximum range limit for the 3D reconstruction method. Additionally, while the article has achieved good results in trajectory tracking, it does not mention the impact on the larger cumulative error caused by the larger reconstruct range.

SPECIFIC NOTES

  1. The authors should introduce the settings and parameters of the drone used in the experiment, such as frame type, motor type, and specifications and manufacturer of the RGB-D camera. Additionally, an enlarged view of the drone in Figure 1 is necessary.
  2. Line 160 may have a possible typo with "marealny."
  3. Citations should be consistent, with the author name as the subject rather than the citation number, as seen in line 119.
  4. The authors should explain PNM in line 206.
  5. Line 312 has a duplicate "we define."
  6. The authors should explain why the percentage is 96% in line 351.

Round 2

Reviewer 2 Report

Comments and Suggestions for Authors

I am expressing my appreciation for the author's efforts in addressing the suggestions and feedback provided for the manuscript. The revisions have improved the overall quality of the manuscript.

I am sorry for the confusion over my suggestion #3. I would like you to revise acronyms and abbreviations and ensure they are defined the first time they are used in the manuscript. For example, the text does not define DTAM, IMU, and LiDAR correctly. 

Moreover, consider the inclusion of a table for acronyms and abbreviations that are used in the paper and explains the meaning of them. Please refer to the MDPI template for guidance on formatting and positioning the table within the manuscript.

Thank you again for your dedication and hard work.